# Personalized Immunotherapy Treatment Strategies for a Dynamical System of Chronic Myelogenous Leukemia

**DOI:** 10.3390/cancers13092030

**Published:** 2021-04-22

**Authors:** Paul A. Valle, Luis N. Coria, Corina Plata

**Affiliations:** Postgraduate Program in Engineering Sciences, BioMath Research Group, Tecnológico Nacional de México/IT Tijuana, Blvd. Alberto Limón Padilla s/n, Mesa de Otay, Tijuana 22500, Mexico; corina.plata@tectijuana.edu.mx

**Keywords:** leukemia, adaptive T-cell therapy, localizing domain, asymptotic stability, in silico

## Abstract

**Simple Summary:**

As computer performance continues to grow at more affordable costs, mathematical modelling and in silico experimentation begin to play a larger role in understanding cancer evolution. The aim of our work is to formulate a control strategy for the Adaptive Cellular Therapy (ACT) that can fully eradicates the Chronic Myelogenous Leukemia (CML) cells population in a mathematical model describing interactions between naive T cells, effector T cells and CML cancer cells in the circulatory blood system. Mathematical analysis and numerical simulations allow us to conclude that it is possible to design a personalized administration protocol for the ACT in the form of a pulse train with asymmetrical waves and a fixed amplitude to achieve complete CML cancer cells eradication. The amplitude of the impulse on which the treatment is applied is given by an arithmetical combination of the parameters of the system with at least a duty cycle of 45 min/day.

**Abstract:**

This paper is devoted to exploring personalized applications of cellular immunotherapy as a control strategy for the treatment of chronic myelogenous leukemia described by a dynamical system of three first-order ordinary differential equations. The latter was achieved by applying both the Localization of Compact Invariant Sets and Lyapunov’s stability theory. Combination of these two approaches allows us to establish sufficient conditions on the immunotherapy treatment parameter to ensure the complete eradication of the leukemia cancer cells. These conditions are given in terms of the system parameters and by performing several in silico experimentations, we formulated a protocol for the therapy application that completely eradicates the leukemia cancer cells population for different initial tumour concentrations. The formulated protocol does not dangerously increase the effector T cells population. Further, complete eradication is considered when solutions go below a finite critical value below which cancer cells cannot longer persist; i.e., one cancer cell. Numerical simulations are consistent with our analytical results.

## 1. Introduction

At the cellular level, cancer is defined as the uncontrolled growth of malignant cells that were able to break free from their regulatory mechanisms. For a malignant cell to emerge, several mutations need to be accumulated in the DNA sequence of a previously normal cell. When the latter occurs, clonal expansion begins in the form of multiple cell divisions, these descendants can undergo further mutations, increasing the possibility of a more aggressive tumor progression. Therefore, under the right conditions, cancer cells from a primary tumor could acquire the ability to enter into the circulatory system, travel to a different site, and invade surrounding organs and tissues in a process known as metastasis. Although, the great majority of primary tumors in humans are benign, metastatic or secondary tumors are responsible for most of the cancer-related deaths. Cancer cells may arise from almost any part of the body, hence their great diversity, grouping more than 200 types and each one with their own complexities. These types may be classified by the hierarchy as follows: epithelial, nonepithelial and mixed multilineage. In this work, we focus on a particular type of nonepithelial malignancy: Chronic Myelogenous Leukemia (CML).

Leukemia is a type of non-solid tumour, resulting from the clonal expansion of abnormal hematopoietic cells. That is, it starts in the blood-forming cells affecting both the peripheral blood and the bone marrow. Leukemias are classified based on their cell type: myelogenous or lymphocytic; and chronicity: acute or chronic. The latter depends on whether most of the abnormal cells are immature, with a faster growth rate; or mature, characterized by a slower growth rate. Regardless of the case, the lifespan of leukemia cells is longer than normal cells. Hence, they accumulate in the bone marrow and spill over into the peripheral blood, crowding out their functional counterparts such as healthy white blood cells, red blood cells, and platelets. Leukemia patients may eventually die due to infections, nutritional deficiencies, and multiple organ failure [1,2]. Statistics from the World Health Organization indicate that 474,519 new cases of leukemia and 311,594 deaths, including both sexes in all groups of ages, were registered worldwide in 2020 [3].

In the particular case of CML, this disease occurs mostly in adults, and very rarely in children. CML can be defined as a myeloproliferative neoplasm predominantly composed of proliferating granulocytes that did not mature completely [4]. Therefore, these cells do not have the ability to fight pathogens as well as a normal white blood cell would. Concerning CML prognosis, physicians need to know the patient’s age, general health, phase of CML, number of blasts in the blood or bone marrow, and the size of the spleen, in order to plan the best treatment strategy. That is, the type of therapy, dose, and intervals of application. The following six treatments are the standard for patients with CML: targeted therapy, chemotherapy, immunotherapy, high-dose chemotherapy with stem cell transplant, donor lymphocyte infusion, and surgery [5]. Nonetheless, new types of treatment are being tested in clinical trials [6].

Mathematical modelling through first-order Ordinary Differential Equations (ODEs) has been a powerful tool in understanding cancer evolution for nearly fifty years [7,8,9,10]. As computer performance continues to grow at more affordable costs [11], the so-called in silico experimentation begins to play a larger role in understanding cancer biology. Increased computing power enables researchers to formulate and execute more complex mathematical models that could potentially explore several scenarios of tumor progression in both short- and long-term. Further, one may include in these models the constantly evolving survival mechanisms of cancer cells, the immune system response, and the application of cancer therapies such as chemotherapy, immunotherapy, and cancer vaccines, among others. Cell–cell interaction models suggest that intermittent therapy is more beneficial to delay cancer relapse as compared to the standard continuous therapy [10].

Although not all models have been useful for clinical application, one can find the following works concerning mathematical models of anticancer therapies relating analytical results and in silico experimentations with clinical studies. de Pillis et al. [12,13] constructed a mathematical model of cancer chemoimmunotherapy involving the dynamics of tumor cells, specific and non-specific immune cells, chemotherapy and two types of immunotherapy. The authors were able to identify appropriate values for the parameters according to empirical data concerning two human patients. This system is given by a set of ODEs and it is one of the most complete mathematical models of cancer evolution under combined treatment application. Numerical simulations were performed, following recommended dosages from drug manufacturers and clinical studies in both therapies. Kronik et al. [14] formulated a mathematical model that could be individualized by patient-specific parameters to predict outcomes on prostate cancer patients treated with an allogeneic whole-cell cancer vaccine, a type of immunotherapy. In the period of one year, they were able to accurately estimate prostate-specific antigen levels in twelve vaccine-responsive patients. Further, the model describes the direct vaccine effect in immune stimulation against prostate cancer by a set of linear and nonlinear ODEs. Kim et al. [15] developed two mathematical models that explore interactions between drug-sensitive and resistant melanoma cells to facilitate adaptive therapy dosing. They were able to fit patient-specific parameter values for each model and illustrate tumor burden for eight patients. Numerical simulations predicted that adaptive therapy would have delayed time to progression by 6–25 months compared to continuous therapy. Among many others, these works highlight the potential benefits of applying mathematical modelling in cancer clinical trials by fitting values related to patient-specific tumor parameters in order to better design therapy dosing for each individual case.

In this paper, we aim to study CML evolution and its treatment by means of nonlinear system theory and in silico experimentation, we apply the adaptive cellular therapy (ACT) as a control strategy to a system describing the dynamics between naive T cells, effector T cells and CML cancer cells in the circulatory blood system by means of three ODEs. The mathematical model was constructed by Helen Moore and Natasha Li [16] and, as discussed by the authors, main difficulties in modelling CML evolution is related to the scarcity of experimental data that can be used to estimate parameters values. However, they were able to determine 12 parameters for their system, each one with their corresponding range of values.

ACT is a promising approach in the treatment of CML, it is one of the original forms of cancer immunotherapy that illustrates how T-cells can recognize and eliminate malignant cells. This therapy requires patients to be infused with a large number of autologous or allogeneic T-cells that have undergone ex vivo selection and modification, expecting them to target leukemia antigens with minimal impact on normal tissues [17,18]. Hence, our interest in investigating both the short- and long-term effects of this treatment over CML evolution and immune response on the system proposed by More and Li by means of the Localization of Compact Invariant Sets (LCIS) method [19,20], the stability theory in the sense of Lyapunov, and in silico experimentation.

The remainder of this paper proceeds as follows. In Section 2, we present the mathematical preliminaries concerning the LCIS method and provide references concerning its application on first-order ODEs dynamical systems describing cancer evolution. In Section 3, we describe the CML-Immunotherapy mathematical model, the dynamics of each equation and the description, values, and units of each parameter. In Section 4, we apply the LCIS method to compute the localizing domain of the CML-Immunotherapy system and derived sufficient conditions by means of Lyapunov’s stability theory to ensure CML cancer cells eradication. In Section 5, we perform the in silico experimentation in the form of several numerical simulations to design personalized protocols for the administration of the ACT treatment. We illustrate how this protocol can be implemented for different initial non-solid tumor concentrations and still successfully eradicate the CML cancer cell population. Further, biological implications and interpretation of these results are also discussed. Finally, conclusions are given in Section 6 and an appendix section is provided where the local existence and uniqueness of solutions is discussed.

## 2. Localization of Compact Invariant Sets Method

The LCIS method was proposed by Krishchenko and Starkov in [19,20] to study the short- and long-time dynamics of nonlinear systems of first-order ODEs by computing the so-called localizing domain. During the past few years, this method has been successfully applied to analyze the global dynamics of cell-cell interaction systems concerning cancer evolution [21,22,23,24,25].

The localizing domain is a bounded region in the state space $R^{n}$ where all compact invariant sets of a system are located. Equilibrium points, periodic, homoclinic and heteroclinic orbits, limit cycles and chaotic attractors are examples of compact invariant sets. Let us take an autonomous nonlinear ODEs system of the form $\dot{x}=f\left(x\right)$, where $f\left(x\right)$ is a $C^{\infty }-$differentiable vector function and $x\in R^{n}$ is the state vector. Let $h\left(x\right):R^{n}\to R$ be a $C^{\infty }-$differentiable function, ${h|}_{S}$ denotes the restriction of $h\left(x\right)$ on a set $S\subset R^{n}$. The function $h\left(x\right)$ used in this statement is called localizing and it is assumed that $h\left(x\right)$ is not the first integral of $f\left(x\right)$. S(h) denotes the set $\left\{x\in R^{n}|L_{f}h\left(x\right)=0\right\}$, where $L_{f}h\left(x\right)$ represents the Lie derivative of $f\left(x\right)$ and is given by: $L_{f}h\left(x\right)=\left(\partial h/\partial x\right)f\left(x\right)$. Now, let us define $h_{\inf}=\inf\left\{h\left(x\right)|x\in S\left(h\right)\right\}$ and $h_{\sup}=\sup\left\{h\left(x\right)|x\in S\left(h\right)\right\}$. Then, the General Theorem concerning the localization of all compact invariant sets of a dynamical system establishes the following:

**Theorem** **1.**
*General Theorem. See Section 2 in [20]. Each compact invariant set *Γ* of $\dot{x}=f\left(x\right)$ is contained in the localizing domain:*
$$K\left(h\right)=\left\{h_{\inf}\leq h\left(x\right)\leq h_{\sup}\right\}.$$


Localizing functions are selected by a heuristic process, this means that one may need to analyze several functions in order to find a proper set that will allow us to fulfill the General Theorem. If one considers the location of all compact invariant sets inside the domain $U\subset R^{n},$ then the set $K\left(h\right)\;\cap \;U$ may be formulated. It is evident that if all compact invariant sets are located in the sets $K\left(h_{i}\right)$ and $K\left(h_{j}\right)$, with $K\left(h_{i}\right),K\left(h_{j}\right)\;\subset \;R^{n}$, then they are located in the set $K\left(h_{i}\right)\;\cap \;K\left(h_{j}\right)$ as well. Furthermore, a refinement of the localizing domain $K\left(h\right)$ is realized with help of the Iterative Theorem stated as follows:

**Theorem** **2.**
*Iterative Theorem. See Section 2 in [20]. Let $h_{m}\left(x\right),m=0,1,2,\cdots$ be a sequence of $C^{\infty }-$differentiable functions. Sets*
$$K_{0}=K\left(h_{0}\right),\;K_{m}=K_{m-1}\cap K_{m-1,m},\;m>0,$$
*with*
$$\begin{array}{c} K_{m-1,m}=\left\{x:h_{m,\inf}\leq h_{m}\left(x\right)\leq h_{m,\sup}\right\},\\ h_{m,\sup}=\underset{S\left(h_{m}\right)\cap K_{m-1}}{\sup}h_{m}\left(x\right),\\ h_{m,\inf}=\underset{S\left(h_{m}\right)\cap K_{m-1}}{\inf}h_{m}\left(x\right), \end{array}$$
*contain any compact invariant set of the system $\dot{x}=f\left(x\right)$ and*
$$K_{0}⊇K_{1}⊇\cdots ⊇K_{m}⊇\cdots \;.$$


## 3. The CML-Immunotherapy Mathematical Model

The mathematical model for CML evolution was formulated by Moore and Li by means of three first-order ODEs that describe the interactions between naive T cells $\left[T_{n}\left(t\right)\right]$, effector T cells $\left[T_{e}\left(t\right)\right]$ and leukemia cancer cells $\left[C\left(t\right)\right]$ in the circulatory blood system [16]. In the latter, authors compute equilibrium points and study their local stability by linearization of the system. They also perform a numerical analysis of the relationships between parameters and its influence in the leukemia cancer cells population. The latter allows them to conclude on the biological relevance, validation of the model and how it could be used to solve the optimal drug dosing problem for leukemia patients. Hence, we use their mathematical model to solve the cancer cells eradication problem by considering a cellular immunotherapy treatment application. The CML-Immunotherapy mathematical model is given as follows.
(1)$$\begin{array}{ccc} {\dot{T}}_{n} & = & s_{n}-d_{n}T_{n}-k_{n}\frac{C}{C+\eta }T_{n}, \end{array}$$
(2)$$\begin{array}{ccc} {\dot{T}}_{e} & = & \alpha_{n}k_{n}\frac{C}{C+\eta }T_{n}+\alpha_{e}\frac{C}{C+\eta }T_{e}-d_{e}T_{e}-\gamma_{e}CT_{e}+\phi_{i}, \end{array}$$
(3)$$\begin{array}{ccc} \dot{C} & = & r_{c}C\ln\left(\frac{C_{\max}}{C}\right)-d_{c}C-\gamma_{c}CT_{e}. \end{array}$$

These equations describe the time-evolution of each cell population in the blood compartment and their interactions are discussed below.

The dynamics of naive T cells, both specific and non-specific to CML cancer cells, are described by Equation (Equation 1). Naive T cells are considered to enter the blood system at a constant rate $s_{n}$ with a natural death rate of $d_{n}$ in the absence of a proper stimulus, e.g., the presence of CML cancer cells. The Michaelis-Menten term represents the differentiation of naive T cells into specific effector cells due to encounters with the CML antigen in the lymph nodes at a constant rate $k_{n}$ and a half-saturation of η. Further, the Michaelis-Menten term considers both activation and energy as contributors to the loss term; naive T cells can become anergic if they encounter the CML antigen without co-stimulators.

The effector T cells specific to CML cancer cells that have differentiated from naive T cells are represented by Equation (2). The first term is due to activation encounters between naive T cells and APC presenting peptides from CML antigen. It is assumed that a total of $\alpha_{n}$ naive T cell successfully converts into effector T cells specific to CML cancer cells. The second term is a recruitment term of effector T cells due to their encounter with cancer cells, it is assumed that a proportion $\alpha_{e}$ of effector cells will recruit other immune cells to aid in killing CML cells. The third term represents the natural death of effector T cells with a coefficient of $d_{e}$. The fourth term represents the loss of effector T cells at a rate $\gamma_{e}$ due to activation-induced cell death from leukemia cancer cells. This process is described by the law of mass action as these encounters occur in the blood and there is no saturation effect such as in the lymph nodes. Lastly, $\phi_{i}$ is the adaptive T-cell therapy, i.e., this term represents the infused enhanced T-lymphocytes into the patient.

The growth and evolution of CML cancer cells are given by Equation (3). The first term represents tumor growth in the form of a Gompertz law which is considered to be the best fit for leukemic cancers, i.e., non-solid tumors. The CML cancer cells proliferation rate is given by $r_{c}$ with a maximum carrying capacity of $C_{\max}$, this limit could be decreased by considering the natural death of these cells which is given by the second term with a constant rate of $d_{c}$. The loss of cancer cells due to their encounter with effector T cells is described by the law of mass action with a coefficient of $\gamma_{c}$. This last term is directly influenced by the cellular immunotherapy treatment concentration in the blood system.

The description, range of values, and units of each parameter of the CML-Immunotherapy mathematical model (Equation 1)–(3) is shown in Table 1 and were retrieved from [16].

Furthermore, the CML-Immunotherapy system (Equation 1)–(3) fulfills the positivity property established by De Leenheer and Aeyels, see Section II.A in [26]. The latter implies that given nonnegative initial conditions, all solutions will have nonnegative real values for all $t\in \left[0,\infty \right)$. Hence, any semi-trajectory of the system is going to be positively forward invariant in the nonnegative octant $R_{+,0}^{3}$ and all dynamics of the system are located in the following domain:$$R_{+,0}^{3}=\left\{T_{n}\left(t\right)>0,T_{e}\left(t\right),C\left(t\right)\geq 0\right\},$$
and it is important to note that given the constant influx $\left(s_{n}\right)$ of naive T cells $\left[T_{n}\left(t\right)\right]$, this population will persist $\left[T_{n}\left(t\right)>0\right]$ as the natural proliferation rate of these cells is higher than its natural death rate $\left(d_{n}\right)$ and differentiation rate into T cells $\left(k_{n}\right)$, i.e., $s_{n}>d_{n}+k_{n}$. Nonetheless, both effector T cells and CML cells populations may be zero; the first one may be zero in the absence of immune stimulation or complete immune suppression, while the other if both the immune response and therapy are successful. Further, local existence and uniqueness of solutions regarding the CML-Immunotherapy system (Equation 1)–(3) is discussed in Appendix A.

## 4. Results

In this section, we will demonstrate how to apply the LCIS method and Lyapunov’s stability theory to derive sufficient conditions on the immunotherapy treatment to ensure the complete eradication of the CML cancer cells population described by system (Equation 1)–(3). These conditions are given by inequalities in terms of the system parameters and are going to be applied with in silico experimentations to formulate a therapy administration protocol in a further section.

### 4.1. Localizing Domain

In order to determine all bounds for a compact localizing domain to the CML-Immunotherapy system we explore four localizing functions. First, let us determine an upper bound for the concentration of naive T cells. Hence, the first localizing function is given by
$$h_{1}=T_{n},$$
and its Lie derivative is as follows
$$L_{f}h_{1}=s_{n}-d_{n}T_{n}-k_{n}T_{n}\left(\frac{C}{C+\eta }\right),$$
hence, set $S\left(h_{1}\right)=\left\{L_{f}h_{1}=0\right\}$ may be written as shown below
$$S\left(h_{1}\right)=\left\{d_{n}T_{n}=s_{n}-k_{n}T_{n}\left(\frac{C}{C+\eta }\right)\right\},$$
now, one can discard the negative term on the right-hand side and estimate the next upper bound for the naive T cells
$$K_{1}\left(h_{1}\right)=\left\{T_{n}\left(t\right)\leq T_{n\sup}=\frac{s_{n}}{d_{n}}\right\}.$$

The lower bound may be computed as well from set $S(h_{1})$ when rewriting it as follows
$$S\left(h_{1}\right)=\left\{\left(d_{n}+k_{n}\right)T_{n}=s_{n}+\frac{k_{n}\eta }{C+\eta }T_{n}\right\},$$
thus, the Michaelis–Menten term on the right-hand side is discarded and the lower bound is estimated
$$K_{2}\left(h_{1}\right)=\left\{T_{n}\left(t\right)\geq T_{n\inf}=\frac{s_{n}}{d_{n}+k_{n}}\right\}.$$

From the latter, the next result concerning the ultimate bounds for the naive T cells population is established in the following set
$$K_{T_{n}}=\left\{T_{n\inf}=\frac{s_{n}}{d_{n}+k_{n}}\leq T_{n}\left(t\right)\leq T_{n\sup}=\frac{s_{n}}{d_{n}}\right\}.$$

Now, let us consider the following localizing function to compute an upper bound for the CML cancer cells
$$h_{2}=C,$$
hence, the Lie derivative is given by
$$L_{f}h_{2}=r_{c}C\ln\left(\frac{C_{\max}}{C}\right)-d_{c}C-\gamma_{c}CT_{e},$$
and set $S\left(h_{2}\right)=\left\{L_{f}h_{2}=0\right\}$ is written as shown below
$$S\left(h_{2}\right)=\left\{\left(r_{c}\ln C_{\max}-r_{c}\ln C-d_{c}-\gamma_{c}T_{e}\right)C=0\right\},$$
thus, the following two solutions can be found
$$S\left(h_{2}\right)=\left\{\ln C=\ln C_{\max}-\frac{d_{c}}{r_{c}}-\frac{\gamma_{c}}{r_{c}}T_{e}\right\}\cup \left\{C=0\right\},$$
from the latter, the lower bound is determined and a preliminary upper bound for any solution $C\left(t\right)$ is estimated as follows
$$K\left(h_{2}\right)=\left\{0\leq C\left(t\right)\leq C_{\max}e^{-d_{c}/r_{c}}\right\},$$
this upper bound will be improved below by applying the Iterative Theorem when considering both the immune response and the immunotherapy treatment.

The third localizing function is intended to study the dynamics of the effector T cells population. The function is given by
$$h_{3}=T_{e},$$
and the Lie derivative is computed below
$$L_{f}h_{3}=\alpha_{n}k_{n}T_{n}\left(\frac{C}{C+\eta }\right)+\alpha_{e}T_{e}\left(\frac{C}{C+\eta }\right)-d_{e}T_{e}-\gamma_{e}CT_{e}+\phi_{i},$$
thus, set $S\left(h_{3}\right)=\left\{L_{f}h_{3}=0\right\}$ is presented as follows
$$S\left(h_{3}\right)=\left\{T_{e}\left(d_{e}+\gamma_{e}C\right)=\alpha_{n}k_{n}T_{n}\left(\frac{C}{C+\eta }\right)+\alpha_{e}T_{e}\left(\frac{C}{C+\eta }\right)+\phi_{i}\right\},$$
hence, if we discard the Michaelis–Menten terms of the right-hand side to estimate a lower bound, then the following result can be determined
$$S\left(h_{3}\right)\subset \left\{\left(d_{e}+\gamma_{e}C\right)T_{e}\geq \phi_{i}\right\},$$
where the Iterative Theorem is applied when considering immune suppression from CML cancer cells as indicated below
$$S\left(h_{3}\right)\cap K\left(h_{2}\right)\subset \left\{\left(d_{e}+\gamma_{e}C_{\max}e^{-d_{c}/r_{c}}\right)T_{e}\geq \phi_{i}\right\},$$
therefore, the lower bound for the effector T cells population is given by
$$K\left(h_{3}\right)=\left\{T_{e}\left(t\right)\geq T_{e\inf}=\frac{\phi_{i}}{d_{e}+\gamma_{e}C_{\max}e^{-d_{c}/r_{c}}}\right\}.$$

This previous result allows us to improve the upper bound of the CML cancer cells population. Now, both the immune response and the immunotherapy treatment are going to be considered by applying the Iterative Theorem to set $S\left(h_{2}\right)$ as follows
$$S\left(h_{2}\right)\cap K\left(h_{3}\right)\subset \left\{\ln C\leq \ln C_{\max}-\frac{d_{c}}{r_{c}}-\frac{\gamma_{c}}{r_{c}}T_{e\inf}\right\}\cup \left\{C=0\right\},$$
and by taking the exponential form on both sides of the equation we get the following ultimate lower and upper bounds for the leukemia cancer cells population
$$K_{C}=\left\{0\leq C\left(t\right)\leq C_{\sup}=C_{\max}e^{-\left(d_{c}+\gamma_{c}T_{e\inf}\right)/r_{c}}\right\}.$$

Now, to estimate the upper bound for the effector T cells population one can analyze the following localizing function
$$h_{4}=T_{e}+\ln C,$$
the Lie derivative is computed as indicated below
$$\begin{array}{ccc} L_{f}h_{4} & = & \alpha_{n}k_{n}\frac{C}{C+\eta }T_{n}+\alpha_{e}\frac{C}{C+\eta }T_{e}-d_{e}T_{e}-\gamma_{e}CT_{e}\\ & & +\phi_{i}+\frac{1}{C}\left[r_{c}C\ln\left(\frac{C_{\max}}{C}\right)-d_{c}C-\gamma_{c}CT_{e}\right], \end{array}$$
and set $S\left(h_{4}\right)=\left\{L_{f}h_{4}=0\right\}$ is written as follows
$$\begin{array}{ccc} S\left(h_{4}\right) & = & \left\{\Lambda T_{e}=\alpha_{n}k_{n}T_{n}-\frac{\alpha_{n}k_{n}T_{n}\eta }{C+\eta }-\frac{\alpha_{e}T_{e}\eta }{C+\eta }\right.\\ & & \left.-\gamma_{e}CT_{e}+\phi_{i}+r_{c}\ln C_{\max}-r_{c}\ln C-d_{c}\right\}, \end{array}$$
where
(4)$$\Lambda =d_{e}+\gamma_{c}-\alpha_{e}>0,$$
thus, by considering $T_{e}=h_{4}-\ln C$, we get the following
$$\begin{array}{ccc} S\left(h_{4}\right) & = & \left\{\Lambda h_{4}-\Lambda \ln C=\phi_{i}-d_{c}+\alpha_{n}k_{n}T_{n}+r_{c}\ln C_{\max}\right.\\ & & \left.-r_{c}\ln C-\frac{\alpha_{n}k_{n}T_{n}\eta }{C+\eta }-\frac{\alpha_{e}T_{e}\eta }{C+\eta }-\gamma_{e}CT_{e}\right\}, \end{array}$$
and rewrite the latter by assuming that next condition holds
(5)$$0<\Lambda <r_{c},$$
therefore
$$\begin{array}{ccc} S\left(h_{4}\right) & = & \left\{h_{4}=\frac{1}{\Lambda }\left(\phi_{i}-d_{c}+\alpha_{n}k_{n}T_{n}+r_{c}\ln C_{\max}\right)\right.\\ & & \left.-\frac{1}{\Lambda }\left[\left(r_{c}-\Lambda \right)\ln C+\frac{\alpha_{n}k_{n}T_{n}\eta }{C+\eta }+\frac{\alpha_{e}T_{e}\eta }{C+\eta }+\gamma_{e}CT_{e}\right]\right\}, \end{array}$$
and the Iterative Theorem is applied to get the following subset
$$S\left(h_{4}\right)\cap K_{1}\left(h_{1}\right)\subset \left\{h_{4}\leq \frac{1}{\Lambda }\left(\phi_{i}-d_{c}+\alpha_{n}k_{n}T_{n\sup}+r_{c}\ln C_{\max}\right)\right\},$$
from the latter, the next constraint is formulated
(6)$$\phi_{i}-d_{c}+\alpha_{n}k_{n}T_{n\sup}+r_{c}\ln C_{\max}>0,$$
if (Equation 6) is fulfilled, one can conclude the next upper limit for the localizing function $h_{4}$ as follows
$$K\left(h_{4}\right)=\left\{T_{e}\left(t\right)+\ln C\left(t\right)\leq \frac{1}{\Lambda }\left(\phi_{i}-d_{c}+\alpha_{n}k_{n}T_{n\sup}+r_{c}\ln C_{\max}\right)\right\}.$$
Now, the next upper bound for the effector T cells population can be estimated from $K\left(h_{4}\right)$
$$K_{1}\left(h_{4}\right)=\left\{T_{e}\left(t\right)\leq T_{e\sup}=\frac{1}{\Lambda }\left(\phi_{i}-d_{c}+\alpha_{n}k_{n}T_{n\sup}+r_{c}\ln C_{\max}\right)\right\},$$
thus, lower and upper bounds for any nondivergent solutions of $T_{e}\left(t\right)$ to the effector T cells population are given in the next set
$$K_{T_{e}}=\left\{T_{e\inf}=\frac{\phi_{i}}{d_{e}+\gamma_{e}C_{\max}e^{-d_{c}/r_{c}}}\leq T_{e}\left(t\right)\leq T_{e\sup}=\frac{\phi_{i}-d_{c}+\alpha_{n}k_{n}T_{n\sup}+r_{c}\ln C_{\max}}{d_{e}+\gamma_{c}-\alpha_{e}}\right\}.$$

Results shown in this section allow us to formulate the following statement concerning the ultimate bounds of a compact domain located in the nonnegative octant $R_{+,0}^{3}$ for the CML-Immunotherapy system (Equation 1)–(3).

**Theorem** **3.**
*Localizing Domain. If conditions (Equation 4)–(Equation 6) hold, then the CML-Immunotherapy system (Equation 1)–(3) has the following compact localizing domain containing all its compact invariant sets*
$$K_{\Gamma }=K_{T_{n}}\cap K_{C}\cap K_{T_{e}},$$
*where*
$$\begin{array}{ccc} K_{T_{n}} & = & \left\{T_{n\inf}=\frac{s_{n}}{d_{n}+k_{n}}\leq T_{n}\left(t\right)\leq T_{n\sup}=\frac{s_{n}}{d_{n}}\right\},\\ K_{C} & = & \left\{0\leq C\left(t\right)\leq C_{\sup}=C_{\max}e^{-\left(d_{c}+\gamma_{c}T_{e\inf}\right)/r_{c}}\right\},\\ K_{T_{e}} & = & \left\{T_{e\inf}=\frac{\phi_{i}}{d_{e}+\gamma_{e}C_{\max}e^{-d_{c}/r_{c}}}\leq T_{e}\left(t\right)\leq T_{e\sup}=\frac{\phi_{i}-d_{c}+\alpha_{n}k_{n}T_{n\sup}+r_{c}\ln C_{\max}}{d_{e}+\gamma_{c}-\alpha_{e}}\right\}. \end{array}$$


The latter implies that all meaningful dynamics of the system will be forward invariant located within the localizing domain $K_{\Gamma }\subset R_{+,0}^{3}$. These dynamics include both tumor-free and tumor-burden equilibria. Hence, sufficient conditions for asymptotic stability of the tumor-free equilibrium point should be derived to properly design an applicable immunotherapy treatment protocol that could potentially control and take the system to a clinically healthy state, i.e., cancer remission or complete tumour eradication.

### 4.2. Leukemia Cancer Cells Eradication

Now, we will demonstrate how to derive sufficient conditions to ensure the complete leukemia cancer cells eradication in the CML-Immunotherapy system (Equation 1)–(3). First, when nonlinear systems theory is combined with systems biology and in silico experimentation, it can be assumed that there is a final critical value below which cancer cells can no longer persist [27,28]. Taking this into account, there is no biological meaning for any numerical value describing fewer than 1 cell in the domain of the variables of the system. Therefore, this threshold is established to discuss and ensure the complete eradication of the CML cancer cells population by means of the immunotherapy treatment application. The latter allows us to propose the following.

**Assumption** **1.**
*Threshold for CML Cancer Cells Eradication. For any solution $C\left(t\right)$ that goes below the threshold value of *1* cell, it is possible to assume complete eradication of the CML cancer cells described by system (Equation 1)–(3). Hence,*
$$C\left(t\right)=0\;\forall \;C\left(t\right)<1.$$


Furthermore, Moore and Li [16] discuss partial remission when there are fewer than 22,500 CML cancer cells/μL. Results shown in this section are determined by means of Lyapunov’s stability theory (see Theorem 4.2 by Khalil in [29] at Section 4.1) and the Localizing Domain Theorem. Thus, let us exploit the following candidate Lyapunov function
$$h_{5}=C,$$
and compute its derivative as follows
$$L_{f}h_{5}=\left[r_{c}\ln\left(\frac{C_{\max}}{C}\right)-d_{c}-\gamma_{c}T_{e}\right]C.$$
It is evident that $L_{f}h_{5}\left(0\right)=0,$ and in virtue of ${\left.L_{f}h_{5}\right|}_{K_{\Gamma }}$ when considering the Threshold for CML Cancer Cells Eradication, one can formulate the next upper bound to function $L_{f}h_{5}$
$$L_{f}h_{5}\leq \left(r_{c}\ln C_{\max}-d_{c}-\gamma_{c}T_{e_{\inf}}\right)C\leq 0,$$
from the latter, the next constraint is determined
$$r_{c}\ln C_{\max}-d_{c}-\frac{\gamma_{c}\phi_{i}}{d_{e}+\gamma_{e}C_{\max}e^{-d_{c}/r_{c}}}<0,$$
and solved for the immunotherapy treatment as indicated below
(7)$$\phi_{i}>\phi_{\inf}=\frac{\left(r_{c}\ln C_{\max}-d_{c}\right)\left(d_{e}+\gamma_{e}C_{\max}e^{-d_{c}/r_{c}}\right)}{\gamma_{c}}.$$

Therefore, we are able to formulate the following statement regarding the leukemia cancer cells eradication by means of the immunotherapy treatment application.

**Theorem** **4.**
*CML Cancer Cells Eradication. If the concentration of the immunotherapy treatment $\phi_{i}$ fulfills condition (Equation 7), then complete eradication of the CML cancer cells population described by system (Equation 1)–(3) is achieved. Hence,*
$$\lim_{t\to \infty }C\left(t\right)=0.$$


The latter implies asymptotic stability for all solutions $C\left(t\right)$ to the plane C=0. Then, when conditions for the CML Cancer Cells Eradication Theorem are fulfilled, the CML-Immunotherapy system (Equation 1)–(3) is simplified as follows
$$\begin{array}{ccc} {\dot{T}}_{n} & = & s_{n}-d_{n}T_{n},\\ {\dot{T}}_{e} & = & \phi_{i}-d_{e}T_{e}, \end{array}$$
which is an uncoupled linear system with a unique equilibrium point $\left(T_{n}^{*},T_{e}^{*}\right)$ = $\left(s_{n}/d_{n},\phi_{i}/d_{e}\right)$ that is globally asymptotically stable (see Theorem 4.5 by Khalil in [29] at Section 4.3). Additionally, when conditions for the CML Cancer Cells Eradication Theorem are fulfilled, then all solutions of system (Equation 1)–(3) with nonnegative initial conditions will go to the tumour-free equilibrium point
(8)$$\left(T_{n}^{*},T_{e}^{*},C^{*}\right)=\left(\frac{s_{n}}{d_{n}},\frac{\phi_{i}}{d_{e}},0\right),$$
and we come to the following result.

**Theorem** **5.**
*Global Stability. If condition (Equation 7) holds, then the tumour-free equilibrium point (Equation 8) of the CML-Immunotherapy system (Equation 1)–(3) is globally asymptotically stable in $R_{+,0}^{3}$.*


## 5. Discussion and In Silico Experimentation

In this section, we discuss and illustrate our mathematical results concerning the CML cancer cells eradication by an immunotherapy treatment applied as a control strategy. In silico experimentations were performed to formulate an administration protocol that eradicates the CML cancer cells population when considering three different initial non-solid tumour sizes, initial conditions $C\left(0\right)$ are set as ratios of the maximum carrying capacity, $C_{\max}$; a fast-growing tumor rate $\left(r_{c}\right)$; a persistence leukemia cells population with a low death rate $\left(d_{c}\right)$; and an enhanced elimination rate of CML cells by the effector T cells $\left(\gamma_{c}\right)$ due to the therapy. Table 1 shows the specific value used for each parameter in all numerical simulations.

Now, let us discuss conditions for the Localizing Domain Theorem, i.e., (Equation 4)–(Equation 6). According to the values of Table 1, these conditions hold, which implies that all compact invariant sets of the CML-Immunotherapy system (Equation 1)–(3) are located either inside or at the boundaries of the domain $K_{\Gamma }$. Hence, all nondivergent solutions are bounded from above and below. Further, it is important to notice that these conditions were derived from function $h_{4}$. Therefore, in the case these conditions are not fulfilled, both naive T cells $\left[T_{n}\left(t\right)\right]$ and CML cancer cells $\left[C\left(t\right)\right]$ will still have their corresponding lower and upper bounds. Additionally, existence of the lower bound for the effector T cells $\left[T_{e}\left(t\right)\right]$ population is not related to conditions (Equation 4)–(Equation 6), and one can still establish sufficient conditions to ensure the CML cancer cells eradication by means of the immunotherapy treatment application.

Below, by substituting values from Table 1 into condition (Equation 7) $\left(\phi_{i}>\phi_{\inf}\right)$ we get the following
$$\phi_{\inf}=\frac{\left(r_{c}\ln C_{\max}-d_{c}\right)\left(d_{e}+\gamma_{e}C_{\max}e^{-d_{c}/r_{c}}\right)}{\gamma_{c}}=15,244\;\mathrm{cells}/\mu L,$$
and, one may set the final cellular immunotherapy treatment concentration as
$$\phi_{i}=1.01\times \phi_{\inf}=15,396\;\mathrm{cells}/\mu L,$$
which is sufficient to ensure the CML cancer cells eradication and global stability of the tumour-free equilibrium point (Equation 8). The latter is written numerically as follows
$$\left(T_{n}^{*},T_{e}^{*},C^{*}\right)=\left(\frac{s_{n}}{d_{n}},\frac{\phi_{i}}{d_{e}},0\right)=\left(1080,256,596,0\right)\times \;\mathrm{cells}/\mu L.$$

Dynamics of the CML-Immunotherapy system (Equation 1)–(3) concerning results discussed above are illustrated in Figure 1. Initial conditions for both naive T cells and effector T cells are set to $T_{n}\left(0\right)=1510$ cells/μL and $T_{e}\left(0\right)=20$ cells/μL, respectively, as indicated by Moore and Li [16]; these values are applied to the remaining numerical simulations shown in this section. For the leukemia cancer cells concentration we used the maximum carrying capacity as initial condition $\left[C\left(0\right)=C_{\max}\right]$ in order to consider the worst-case scenario of tumour burden.

Results shown in Figure 1 are achieved when immunotherapy treatment is constantly applied, i.e., $\phi_{i}=1.01\times \phi_{\inf}$ ∀ t≥0 as it is shown in the upper panel. The application of the treatment was considered as a constant for the sake of simplicity in the mathematical analysis as has been previously discussed through the years by many authors when modelling the role of immunotherapy in boosting the immune system response to cancer growth, see [9,12,13,30,31,32,33,34,35]. Nonetheless, two issues arise in this case, it is not biologically feasible to constantly applied an immunotherapy treatment to a cancer patient and it is evident that the solution of the effector T cells $T_{e}\left(t\right)$ goes to the value $T_{e}^{*}=\phi_{i}/d_{e}=$ 256,596 cells/μL (a concentration close to that of the the maximum carrying capacity of the CML cancer cells) which could produce adverse events in the patient’s health [36]. Hence, in order to avoid the latter, numerous authors have proposed different treatment strategies to apply therapies such as chemotherapy, radiotherapy and immunotherapy by time intervals or in the form of periodic oscillations, one can see [14,37,38,39,40,41,42,43,44]. Thus, our hypothesis is as follows: Daily applications for a finite period of time of the immunotherapy treatment will decrease the CML cancer cells concentration below a critical threshold under which it is possible to ensure the complete eradication of the disease.

In order to apply the treatment for short intervals of time every day we decided to explore a pulse train with asymmetrical waves and a fixed amplitude of $\phi_{i}=1.01\times \phi_{\inf}$. By performing an in silico experimentation process by means of several numerical simulations where duty cycles were increased by 15min in each iteration we were able to conclude that daily applications of the cellular immunotherapy treatment with at least a duty cycle of 45min/day $\left(3.1\%\right)$ were needed to achieve complete CML cancer cells eradication when considering the maximum carrying capacity as initial condition $\left[C\left(0\right)=C_{\max}\right]$. This strategy is illustrated below in Figure 2, one can see the immunotherapy administration protocol in 1 day in the upper panel (time unit is given in hours), and 1 week in the lower panel (time unit is given in days).

The immunotherapy administration protocol from Figure 2 is applied when considering three different initial tumour concentrations, $C\left(0\right)=C_{\max}$ in Figure 3, $C\left(0\right)=\frac{2}{3}C_{\max}$ in Figure 4, and $C\left(0\right)=\frac{1}{3}C_{\max}$ in Figure 5. Solutions to Equations (Equation 1)–(3) as well as the treatment $\left(\phi_{i}\right)$ are illustrated in each case. It is evident that as the initial tumour density increases, then the immunotherapy treatment should be applied for a longer period of time.

Numerical simulations were set to stop the immunotherapy application when cancer cells go below the positive finite critical value of one cancer cell, as discussed in Assumption 1. Hence, both the treatment and the CML cancer cells population were set to zero, i.e., $\phi_{i}=0$ and $C\left(t\right)=0,$ when $C\left(t\right)<1$. After the CML cancer cells eradication, one can see that solutions to both naive T cells and effector T cells go to their respective equilibrium point when $\phi_{i}=0$. The latter is given by $T_{n}^{*}=s_{n}/d_{n}=1080$ cells/μL and $T_{e}^{*}=\phi_{i}/d_{e}=0$. Further, it is important to notice that the final dynamics of the naive T cells concentration is the average value expected in healthy individuals in steady state [16,45,46,47]. Concerning the maximum concentration of effector T cells, results are as follows: 367 cells/μL in Figure 3, 455 cells/μL in Figure 4, and 371 cells/μL in Figure 5. These values are significantly lower than the 256,596 cells/μL obtained in Figure 1 when the immunotherapy treatment is applied constantly over time.

Now, results concerning the CML cancer cells $\left[C\left(t\right)\right]$ populations for each initial concentration are summarized in Figure 6. These dynamics are illustrated in both linear and logarithmic scales in the upper and lower panels, respectively. In the upper panel one can see the threshold for tumour remission ($C_{Re}=$ 22,500 cells/μL) and the tumour-free plane $\left(C^{*}=0\right)$. Whilst, in the lower panel, both thresholds for tumor remission and tumor eradication $(C_{Th}=1$ cell/μL) are shown. Times when CML cancer cells go below the tumour eradication threshold are as follows: 76.4 days for $C_{1}\left(t\right)$ (initial concentration $C\left(0\right)=\frac{1}{3}C_{\max}$); 148.2 days for $C_{2}\left(t\right)$ (initial concentration $C\left(0\right)=\frac{2}{3}C_{\max}$); and 159.4 days for $C_{3}\left(t\right)$ (initial concentration $C\left(0\right)=C_{\max}$). These results are also indicated in Figure 3, Figure 4 and Figure 5.

When condition (Equation 7) is fulfilled, then the CML cancer cells population will decrease as it is stated by the CML Cancer Cells Eradication Theorem. Therefore, it is necessary to note that immediately after stopping treatment in each iteration, then the leukemia cancer cells begin to grow again, which implies that is imperative for the patient to continue with the daily applications of the treatment until the cancer cells concentration goes below the established threshold $\left(C_{Th}\right)$ of one cancer cell.

We expect our work will benefit a leukemia patient in the sense that the methodology described in this paper could be applied to solve the optimal ACT dosing problem for each particular case in any scenario that may potentially be described by the CML-Immunotherapy system (Equation 1)–(3). The mathematical analysis indicates that only a subset of parameters influence the amount of treatment concentration that should ultimately be applied to the patient in order to fulfill condition (Equation 7) to achieve CML Cancer Cells Eradication from Theorem 4 and, it should be noted that this expression is written as a simple algebraic combination of this subset of parameters. With this condition, an administration protocol that can be individualized for each patient was designed and, as Moore and Li concluded in their research [16], this kind of control strategy may increase the time-period in which a cancer patient remains healthy. Additionally, the cost of the treatment could be constrained to the severity of the disease in each patient, as immunotherapy is often very expensive [48]. We aim to further extend our work to design a control strategy for the application of combined cancer therapies such as chemoimmunotherapy and, if possible, discussed how these strategies could be useful in avoiding cancer cell resistance to the prolonged administration of these treatments.

## 6. Conclusions

Cancer dynamics result in a complex set of events all happening at the same time in the tumor site. These events may trigger cancer cells to evolve with diverse mechanisms that allow them to escape immune response and invade other tissues and organs in the human body. Adaptive cellular immunotherapy is a type of treatment that can help the immune system to better respond against cancer growth. However, open questions still remain, e.g., What is the right dose of the treatment to be applied? and How often should the treatment be applied? Mathematical modelling and nonlinear systems theory combined with in silico experimentation may help us to provide scientifically formulated answers to these important questions related to cancer evolution.

Moore and Li [16] provided some insights concerning local stability of the two main equilibrium points of their system, i.e., the healthy state and one with tumor burden. They determined, under certain conditions on the parameters, that both could be locally asymptotically stable. Further, their numerical analysis regarding relationships between parameters and the dynamics of solutions $C\left(t\right)$ indicate that both the growth rate $\left(r_{c}\right)$ and natural death rate $\left(d_{c}\right)$ of CML cancer cells are the most important parameters to control tumour growth. Nonetheless, conditions for local stability may not always be fulfilled, at least not without a proper treatment that can influence the values of the parameters. However, these values are expected to be different for each individual case. Authors conclude their work by providing recommendations on how the model may be applied to include immune system-boosting treatments and to solve the optimal drug dosing problem. Hence, in this paper, we incorporate the adaptive cellular therapy treatment and demonstrate how to apply the LCIS method and Lyapunov’s stability theory to derive sufficient conditions to ensure the complete eradication of the CML cancer cells population, i.e., global asymptotic stability conditions. Further, extensive in silico experimentations allowed us to formulate a control strategy for the administration of the immunotherapy treatment in diverse scenarios.

Numerical simulations allow us to understand and explore diverse scenarios which are illustrated in Figure 3, Figure 4 and Figure 5. We formulate these cases in terms of the initial tumor concentration, initial conditions are set as ratios of $C_{\max}$. This parameter describes the maximum carrying capacity of the CML cancer cells in the circulatory blood system, as it is shown in Table 1. Hence, we decided to explore the dynamics between cancer growth, the immune response and the effect of the cellular immunotherapy treatment for three initial tumour concentration given as follows: $C_{1}\left(0\right)=\frac{1}{3}C_{\max}$, $C_{2}\left(0\right)=\frac{2}{3}C_{\max}$, and $C_{3}\left(0\right)=C_{\max}$. Results concerning the final dynamics of the leukemia cancer cells in each case are summarized in Figure 6. Further, one can see two important numerical values concerning cancer cells, the first is given by the threshold for tumor remission on CML $(C_{Re}=22,500$ cells/μL), and the second by the threshold for complete tumor eradication $(C_{Th}=1$ cell/μL). The latter is discussed in Assumption 1 as we consider that there is no biological meaning for any numerical value describing fewer than one cancer cell. Further, this assumption allows us to establish conditions to ensure both CML cancer cells eradication and global stability of the tumor-free equilibrium point.

Mathematical and in silico results shown in this paper allow us to conclude that it possible to formulate a control strategy for the cellular immunotherapy administration that can fully eradicate the CML cancer cells population described by system (Equation 1)–(3). Numerical simulations illustrate that the total iterations of the treatment application are directly related to the initial tumor concentration in order to decrease it below the final critical value of one cancer cell. The amplitude of the impulse on which the treatment is applied is given by an arithmetical combination of the parameters of the system, as it is written in condition (Equation 7), i.e., $\phi_{i}>\phi_{\inf}=\left(r_{c}\ln C_{\max}-d_{c}\right)\left(d_{e}+\gamma_{e}C_{\max}e^{-d_{c}/r_{c}}\right)/\gamma_{c}$. In Table 1 one can see the range of values for parameters of the CML-Immunotherapy mathematical model. Hence, the latter implies that therapy doses could be personalized for an individual patient as parameters may have different values for each particular case. 

## Figures and Tables

**Figure 1 cancers-13-02030-f001:**
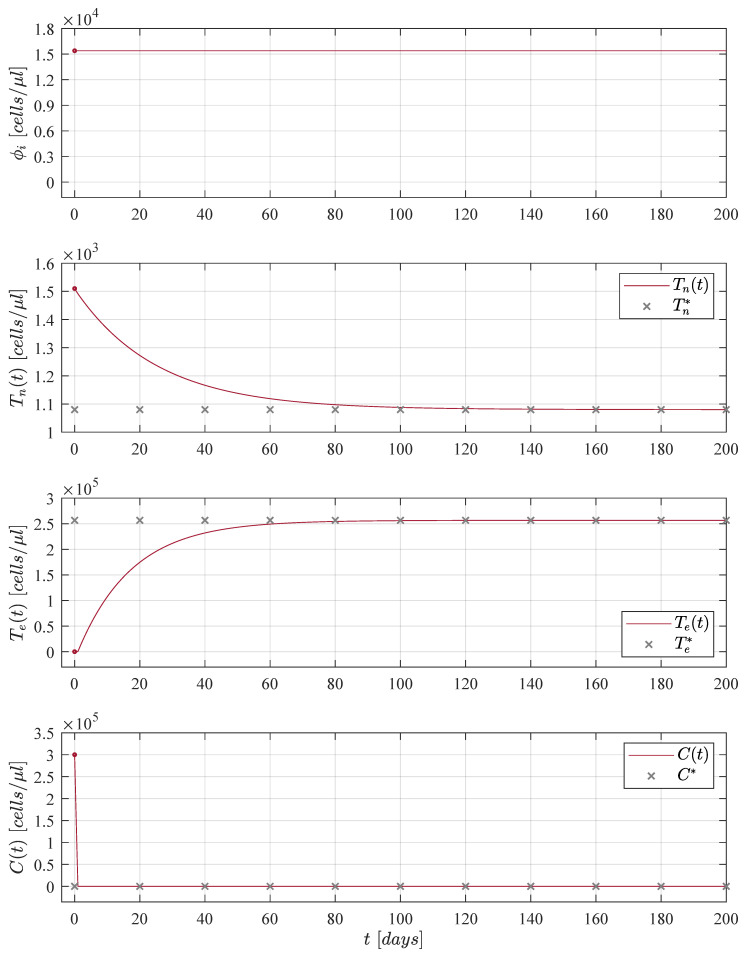
When the cellular immunotherapy treatment parameter fulfills condition (Equation 7), i.e., $\phi_{i}=1.01\times \phi_{\inf}$, all solutions for the CML-Immunotherapy system (Equation 1)–(3) go to the tumour-free equilibrium point (Equation 8) given by $\left(T_{n}^{*},T_{e}^{*},C^{*}\right)=\left(s_{n}/d_{n},\phi_{i}/d_{e},0\right)$. For this particular case, we set the initial tumour density as $C\left(0\right)=C_{\max}=3\times 10^{5}$ cells/μL, as we consider the concentration of leukemia cancer cells at the maximum carrying capacity. Different initial conditions will yield the same result: CML cancer cells eradication and asymptotic stability of the tumour-free equilibrium point.

**Figure 2 cancers-13-02030-f002:**
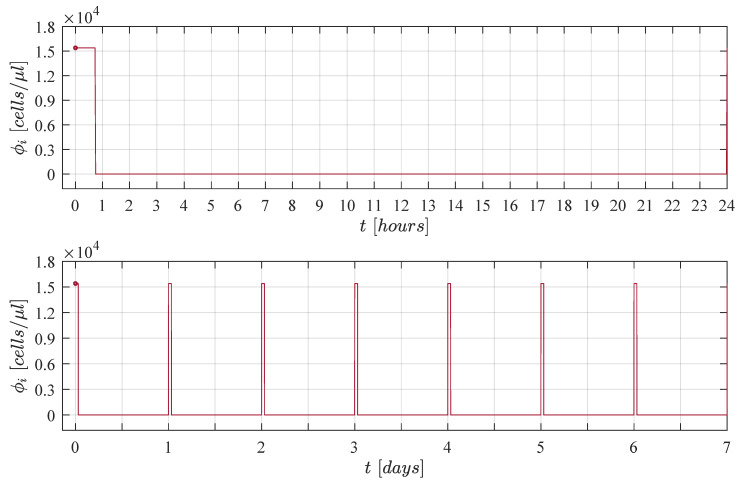
Administration protocol for the immunotherapy treatment. Characteristics are as follows: amplitude of $\phi_{i}=$ 15,396 cells/μL and a duty cycle of 45min/day. Complete CML cancer cells eradication was achieved by means of this strategy for three considered initial non-solid tumour sizes given by $C\left(0\right)=\frac{1}{3}C_{\max}$, $C\left(0\right)=\frac{2}{3}C_{\max}$, and $C\left(0\right)=C_{\max}$.

**Figure 3 cancers-13-02030-f003:**
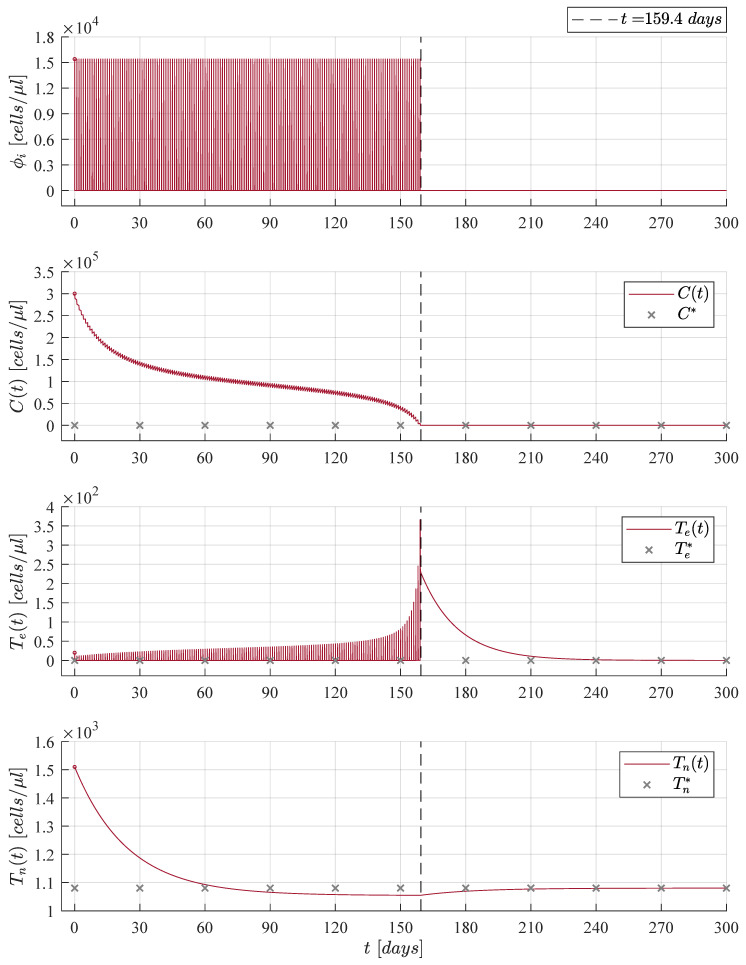
Case 1: The initial tumour size for the CML cancer cells is set as $C\left(0\right)=C_{\max}=3\times 10^{5}$ cells/μL. The cellular immunotherapy $\phi_{i}$ had to be applied 159.4 days to achieve complete leukemia cancer cells eradication.

**Figure 4 cancers-13-02030-f004:**
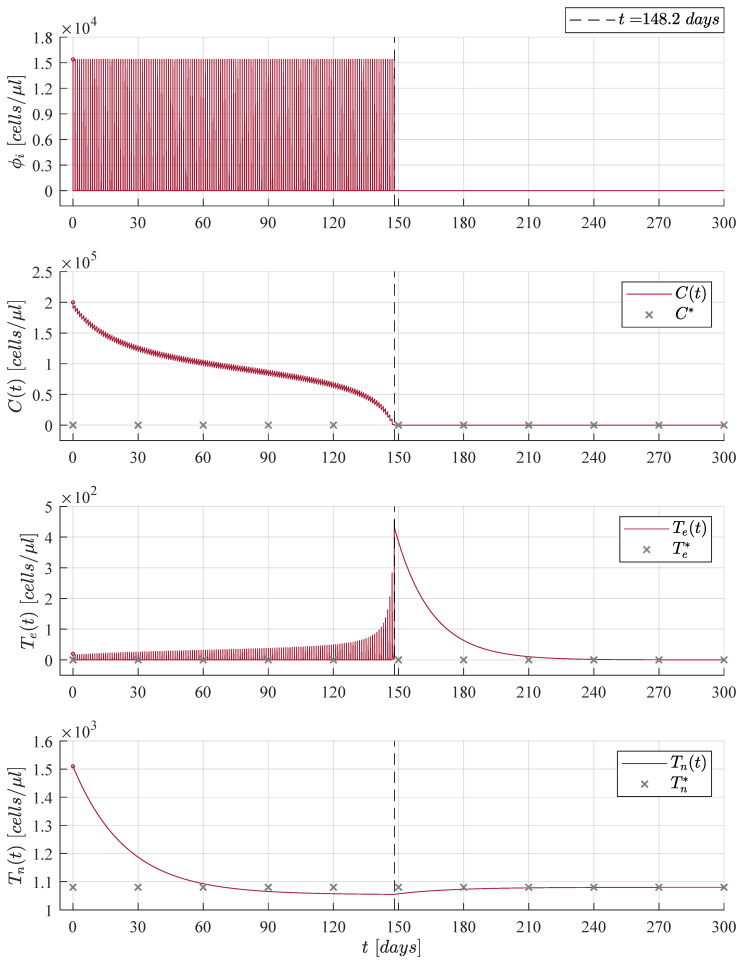
Case 2: The initial tumour size for the CML cancer cells is set as $C\left(0\right)=\frac{2}{3}C_{\max}=2\times 10^{5}$ cells/μL. The cellular immunotherapy $\phi_{i}$ had to be applied 148.2 days to achieve complete leukemia cancer cells eradication.

**Figure 5 cancers-13-02030-f005:**
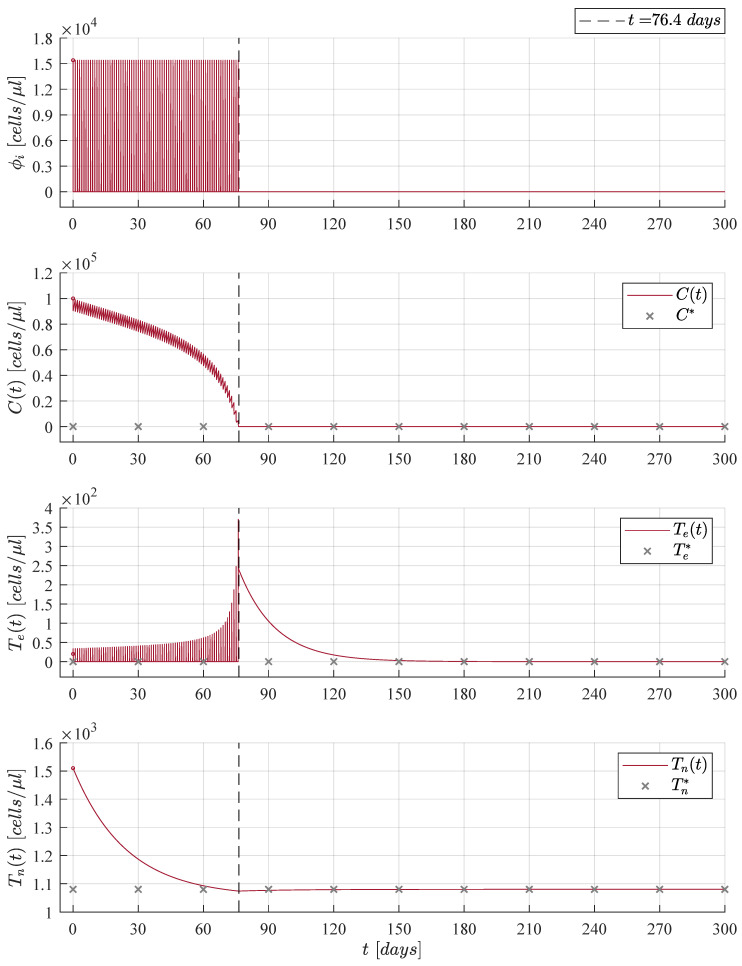
Case 3: The initial tumour size for the CML cancer cells is set as $C\left(0\right)=\frac{1}{3}C_{\max}=1\times 10^{5}$ cells/μL. The cellular immunotherapy $\phi_{i}$ had to be applied 76.4 days to achieve complete leukemia cancer cells eradication.

**Figure 6 cancers-13-02030-f006:**
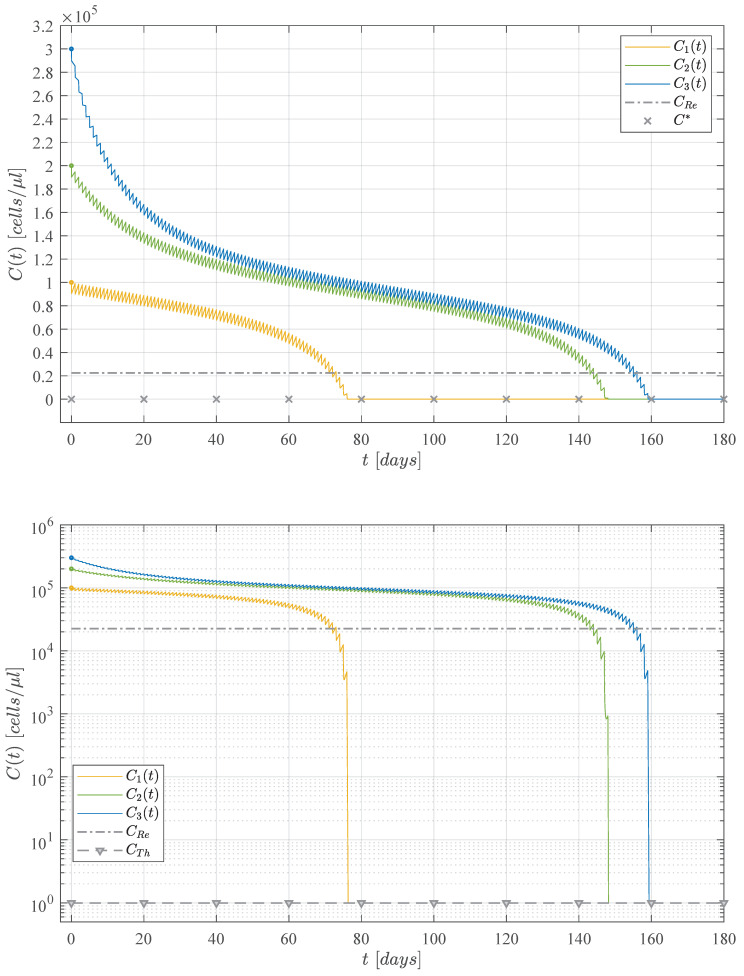
Dynamics of solutions of the CML cancer cells population $\left[C\left(t\right)\right]$ when considering three initial tumour sizes: $C_{1}\left(t\right)$ with $C\left(0\right)=\frac{1}{3}C_{\max}=1\times 10^{5}$ cells/μL, $C_{2}\left(t\right)$ with $C\left(0\right)=\frac{2}{3}C_{\max}=2\times 10^{5}$ cells/μL, and $C_{3}\left(t\right)$ with $C\left(0\right)=C_{\max}=3\times 10^{5}$ cells/μL. Regardless of the initial tumour concentration, complete leukemia cells eradication is achieved in all cases. The immunotherapy administration protocol from Figure 2 is applied to each case, i.e., daily applications of the treatment with a duty cycle of 45min/day $\left(3.1\%\right)$ and a fixed amplitude of $\phi_{i}=1.01\times \phi_{\inf}=$ 15,396 cells/μL.

**Table 1 cancers-13-02030-t001:** Parameter information for the CML-immunotherapy dynamical system.

Parameter	Description	Value	Range	Units
$s_{n}$	Natural proliferation of naive T cells	43.20	$\left(0,50\right]$	cells/(μL × day)
$d_{n}$	Natural death rate of naive T cells	0.040	$\left(0,0.1\right]$	day${}^{-1}$
$k_{n}$	Differentiation rate of naive T cells into effector	0.001	$\left(0,0.1\right]$	day${}^{-1}$
	T cells due to their interaction with CML cells			
η	Half saturation term of T cells recruitment	100	$\left(0,1000\right]$	cells/μL
$\alpha_{n}$	Proliferation rate of effector T cells	0.410	$\left(0,1\right]$	
$\alpha_{e}$	Recruitment rate of effector T cells by CML cells	0.100	$\left(0,1\right]$	day${}^{-1}$
$d_{e}$	Natural death rate of effector T cells	0.060	$\left(0,0.5\right]$	day${}^{-1}$
$\gamma_{e}$	Inactivation rate of effector T cells by CML cells	0.005	$\left(0,0.1\right]$	μL/(cells × day)
$\phi_{i}$	Adaptive T-cell therapy	to be estimated		cells/(μL × day)
$r_{c}$	Growth rate of CML cells	0.100	$\left(0,0.5\right]$	day${}^{-1}$
$C_{\max}$	Maximum carrying capacity of the CML cells	$3\times 10^{5}$	$\left[1.5,4\right]\times 10^{5}$	cells/μL
$d_{c}$	Natural death rate of the CML cells	0.020	$\left[0,0.8\right]$	day${}^{-1}$
$\gamma_{c}$	Elimination of CML cells by effector T cells	0.100	$\left(0,0.1\right]$	μL/(cells × day)

## Data Availability

Data is contained within the article.

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
