# Peer review of "Personalized Immunotherapy Treatment Strategies for a Dynamical System of Chronic Myelogenous Leukemia"

_cancers, 2021, doi:10.3390/cancers13092030_

Round 1

Reviewer 1 Report

In this article, the authors study a mathematical model focusing in Chronic Myelogeneous Leukemia (CML), a particular type of Leukemia, which was con- structed by More and Li, using three nonlinear differential equations.

The authors investigate long terms effect of the treatment, adaptative cel- lular therapy (ACT), over CML evolution. To do that, they apply first the Localization of Compact Invariant Sets methods to define the localizing do- main. Then, they use classical tools from the dynamical systems theory, as Lyapunov’s direct method to study the stability of the system and to obtain sufficient conditions to ensure the eradication of the tumor. Finally, they per- form numerical simulations to check the theoretical results.

Although these techniques from dynamical systems are known and the au- thors have used them in other articles, the article is interesting for researchers dedicated to mathematical oncology.

Overall, I recommend publication of this article. Some recommendations in case the authors want to follow them to improve the text:

  • Existence and uniqueness of solutions of the system of differential equation is not proved in the article. I would suggest to the authors write such proofs in the article and to formulate a theorem or proposition with these results.

  • It could be interesting to compare the results of this article with the results of Moore and Li article in a section, explaining the differences obtained.

Reviewer 2 Report

The paper is overall well written. It is suggested that possible ways in which this approach could be of clinical use be discussed (e.g., in the Discussion). As it stands, it is unclear how the work could ultimately be used for the benefit of patients.  This would entail a minor revision of the manuscript.

Author Response

Reviewer #2:

The paper is overall well written. It is suggested that possible ways in which this approach couldbe of clinical use be discussed (e.g., in the Discussion). As it stands, it is unclear how the work couldultimately be used for the beneÖt of patients. This would entail a minor revision of the manuscript.

Authors are deeply grateful for your comments to our manuscript. We appreciatethe time and dedication to our work. Our response to your recommendation is as follows:

Response: As indicated by the reviewer, we included the following paragraph at the end of theDiscussion section (Page 20, lines 344ñ359.):We expect our work will beneÖt a leukemia patient in the sense that the methodology describedin this paper could be applied to solve the optimal ACT dosing problem for each particular casein any scenario that may potentially be describe by the CML-Immunotherapy system (1)ñ(3). Themathematical analysis indicates that only a subset parameters ináuence the amount of treatmentconcentration that should ultimately be applied to the patient in order to fulÖll condition (7) toachieve CML Cancer Cells Eradication from Theorem 4 and, it should be noted that this expressionis written as a simple algebraic combination of this subset of parameters. With this condition, anadministration protocol that can be individualized for each patient was designed and, as Moore andLi concluded in their research [16], this kind of control strategies may increase the time-period inwhich a cancer patient remains healthy. Additionally, the cost of the treatment could be constrainedto the severity of the disease in each patient, as immunotherapy is often very expensive [48]. Weaim to further extend our work to design a control strategy for the application of combined cancertherapies such as chemoimmunotherapy and, if possible, discussed how these strategies could beuseful in avoiding cancer cells resistance to the prolong administration of these treatments.
